# Improved Uniformity of TaO_x_-Based Resistive Switching Memory Device by Inserting Thin SiO_2_ Layer for Neuromorphic System

**DOI:** 10.3390/ma16186136

**Published:** 2023-09-09

**Authors:** Dongyeol Ju, Sunghun Kim, Junwon Jang, Sungjun Kim

**Affiliations:** Division of Electronics and Electrical Engineering, Dongguk University, Seoul 04620, Republic of Korea

**Keywords:** RRAM, bilayer, neuromorphic system, synaptic plasticity, spike-timing-dependent plasticity

## Abstract

RRAM devices operating based on the creation of conductive filaments via the migration of oxygen vacancies are widely studied as promising candidates for next-generation memory devices due to their superior memory characteristics. However, the issues of variation in the resistance state and operating voltage remain key issues that must be addressed. In this study, we propose a TaO_x_/SiO_2_ bilayer device, where the inserted SiO_2_ layer localizes the conductive path, improving uniformity during cycle-to-cycle endurance and retention. Transmission electron microscopy (TEM) and X-ray photoelectron spectroscopy (XPS) confirm the device structure and chemical properties. In addition, various electric pulses are used to investigate the neuromorphic system properties of the device, revealing its good potential for future memory device applications.

## 1. Introduction

Resistive random-access memory (RRAM), as a next-generation non-volatile memory (NVM), is being investigated using a variety of methods to address the issues that the present technology has encountered. RRAM is gaining popularity among NVM technologies due to its features such as fast switching speed, low power consumption, great scalability, long retention, and simple structure [1,2,3,4,5,6,7,8]. The structure of RRAM is composed of an insulating layer sandwiched between the top electrode (TE) and bottom electrode (BE), having a two-terminal structure that is capable of easy fabrication [9]. Furthermore, valence change mechanisms (VCM), which are based on metal oxides, are well suited for neuromorphic applications due to their low-energy operation, good CMOS scalability, and ability to acquire multiple states [10]. The resistive switching occurs between the high-resistance state (HRS) and the low-resistance state (LRS) under applied bias. Various binary metal oxides have been researched to seek their resistive switching (RS) phenomenon for use in the insulating layer of RRAM. Examples of these oxides are ZnO [11,12], AlO_x_ [13,14], TiO_2_ [15], TaO_x_ [16], CuO [17], HfO_x_ [18], and ZrO_x_ [19,20]. Among the above metal oxides, TaO_x_ is known as the leading oxide material that shows outstanding memory characteristics such as high endurance (>10^10^), fast switching speed (<1 ns), and good scalability (<30 nm) [21,22,23,24,25]. Moreover, some researchers reported that the TaO_x_ layer can create oxygen vacancies easily, indicating that it has benefits in forming strong conductive filaments, further supporting its non-volatile memory device application [26,27,28]. However, several issues such as uniformity in operating voltage and resistance state still exist. The cycle-to-cycle and device-to-device are critical considerations in evaluating RRAM devices since the RS phenomenon of oxide materials occurs as a result of the development and rupture of conductive filaments created by the migration of randomly distributed defects [29,30,31]. Several methods exist to improve filamentary uniformity: ion doping [32], bilayer structure [33], and defective circuit breakers [34]. Many reports on bilayer structures have been conducted because of their efficient and simply executable ways of generating stable and controllable resistive switching behavior [35,36,37]. 

Several studies reported the usage of a SiO_2_ thin layer to improve device characteristics [38]. In a study by Wang et al., they inserted a 1~3 nm thick SiO_2_ layer in a TaO_x_-based VCM device, where the thin layer was used as a diffusion limiting layer (DLL). It was noted that the inserted SiO_2_ layer prevents the hard breakdown of the TaO_x_ layer, improving device uniformity [39]. Furthermore, Yu et al. claimed that by incorporating a 6 nm thick SiO_2_ layer on a TaON film, they were able to enhance operational power while limiting leakage current flow due to the insulating properties of the SiO_2_ layer to minimize soft breakdown [40]. 

In this study, we focus on the bipolar resistive switching characteristic and performance improvement by inserting a thin SiO_2_ layer. Compared to the single-layer Pt/TaO_x_ (3 nm)/TaN device, the bilayer Pt/TaO_x_ (3 nm)/SiO_2_ (2 nm)/TaN device shows improved uniformity of resistance states and a unique reset process. Also, the multi-level cell (MLC) characteristics that are important in improving high data storage are investigated by moderating the reset voltage [41]. Furthermore, to implement neuromorphic applications, the pulses for potentiation and depression are applied, and their results are used for the MNIST pattern recognition. Finally, STDP where top and bottom electrodes mimic the human brain’s pre- and post-spikes are emulated to confirm the ability to mimic the human brain [42].

## 2. Experiments

A 100 nm thick TaN bottom electrode was deposited on top of a Si/SiO_2_ substrate using a pure TaN target with an Ar flow rate of 20 sccm. To eliminate surface contaminants, ultrasonic cleaning with acetone, isopropyl alcohol, and DI water was performed sequentially. After cleaning the bottom electrode surface, a 2 nm thick SiO_2_ was deposited using low-pressure chemical vapor deposition (LPCVD) by reacting dichlorosilane (DCS, SiCl_2_H_2_, 40 sccm) with N_2_O (160 sccm) at 785 °C. After the pre-SiO_2_ insulating layer deposition, a 3 nm thick TaO_x_ was deposited on both single and bilayer devices via DC sputtering. The working pressure was 5 mTorr with a gas mixture of Ar (20 sccm) and O_2_ (6 sccm). Using a shadow mask, circular cell patterns with a diameter of 100 μm were prepared. Finally, a Pt top electrode was achieved using a commercial Pt target (99.99% purity) under gas pressure of 3 mTorr and 20 sccm Ar gas. The Keithley 4200 SCS semiconductor parameter analyzer in DC mode and the 4225-PMU ultrafast current-voltage (I-V) pulse module in pulse mode were used to investigate Pt/TaO_x_/TaN and Pt/TaO_x_/SiO_2_/TaN’s electrical characteristics. At room temperature, bias was provided to the top electrode (Pt), while the bottom electrode (TaN) was grounded. A field emission transmission electron microscope (JEOL JEMF200, Tokyo, Japan) and X-ray photoelectron spectroscopy (XPS) were used to validate the device’s properties.

## 3. Results and Discussion

Figure 1 depicts the electrical characteristics of the devices under the DC sweep method. Figure 1a shows the forming process of Pt/TaO_x_/TaN and Pt/TaO_x_/SiO_2_/TaN devices.

Both devices need a soft breakdown process for the device to convert from their initial high-resistance state (HRS) to a low-resistance state (LRS). The Pt/TaO_x_/SiO_2_/TaN device needs more voltage compared to the TaO_x_ single-layer device for the forming process due to the existence of an additional SiO_2_ insulating layer. I–V curves of both devices are shown in Figure 1b, where the same compliance current (1 mA) and the voltages of 2 V and −2 V are applied for reset and set processes, respectively. The window of the Pt/TaO_x_/SiO_2_/TaN device is about ~5 is smaller than that of the Pt/TaO_x_/TaN, which is about ~33. However, when a consequent voltage sweep is applied, the Pt/TaO_x_/SiO_2_/TaN device shows better uniformity, as shown in Figure 1c,d. For the 100-cycle endurance and 10^4^ s retention tests, a read voltage (V_read_) of 0.3 V is used for both devices. The TaO_x_ single-layer device shows high variance at both HRS and LRS, but the TaO_x_/SiO_2_ bilayer device shows a smaller variation of its resistance state. 

To further investigate the effect of inserting a thin SiO_2_ layer, cell-to-cell variance during DC cycles is observed. A total of 10 randomly selected cells are chosen to identify their uniformity during 20 operating cycles. Each cell is applied with the same set and reset voltages of 2 and −2 V with a V_read_ of 0.3 V. As shown in Figure 2a,b, a huge fluctuation is observed in the Pt/TaO_x_/TaN device, disapproving both cycle-to-cycle and cell-to-cell uniformity to be used as a neuromorphic application. The variation in endurance and retention may have occurred due to the hard breakdown of the TaO_x_ single-layer film; in a study by Wang et al., it was reported that during the filament formation of TaO_x_, a strong voltage drop occurs where the conductive filament is not yet formed, affecting device performance. On the other hand, when a thin SiO_2_ layer is added, the inserted layer acts as a breakdown barrier, improving device performance [39].

However, the Pt/TaO_x_/SiO_2_/TaN device shows improved uniformity achieved with a smaller variance. Additionally, as illustrated in Figure 2c, cell-to-cell variability decreases in both HRS and LRS, proving the improved uniformity achieved by inserting a SiO_2_ layer. Figure 3a is a schematic representation of a Pt/TaO_x_/SiO_2_/TaN device.

The device structure is investigated using TEM as shown in Figure 3b. The 100 nm thick Pt and TaN layers are observed with insulating layers sandwiched between them. The thickness of the insulating layers is 3 nm for the TaO_x_ layer and 2 nm for the SiO_2_ layer.

To clarify the existence of each insulating layer, XPS spectra are investigated, as shown in Figure 4. All XPS spectra are calibrated using a C1s spectrum centered at 284.6 eV.

Figure 4a,b show the spectrum data of the first insulating layer, TaO_x_, at an etch time of 3 s. As shown in Figure 4a, two peaks of Ta4f_7/2_ and Ta4f_5/2_ are shown in the Ta 4f spectra, located at the binding energy of about 22.87 eV and 25.21 eV, respectively. Also, in Figure 4b, the O 1s peak exists at 530.9 eV, indicating the TaO_x_ layer. Furthermore, at an etch time of 12 s, a second insulating layer of SiO_2_ is detected in Figure 4c,d. Three peaks of Si^2+^, Si^3+^, and Si^4+^ are located at about 100.98 eV, 102.65 eV, and 103.9 eV. Moreover, the O 1s spectra are depicted in Figure 4d at 531.6 eV for the Si–O bond.

The initial state of Pt/TaO_x_/SiO_2_/TaN is shown in Figure 5a.

Next, the conduction mechanism of the Pt/TaO_x_/SiO_2_/TaN device is shown in Figure 5. The mechanism of filament formation in the TaO_x_/SiO_2_ device has been described in several previous studies [38,39]. The initial state of Pt/TaO_x_/SiO_2_/TaN is shown in Figure 5a. When a negative voltage is applied to the Pt top electrode, oxygen ions (O^2−^) migrate toward the bottom electrode, leaving oxygen vacancies (V_o_^+^). Left oxygen vacancies accumulate and create a conductive filament, which grows from the top electrode and TaO_x_ layer interface. Due to the major drop in electric potential in SiO_2_ film, high electric fields are applied to this layer, creating a localized conducting path in the SiO_2_ film [39]. Then, as shown in Figure 5b, a conducting filament connects the top and bottom electrodes, changing the resistance state of the device from HRS to LRS and turning the device “on”. Moreover, due to the diffusion-limiting role of the SiO_2_ layer, the conductive filament is formed and ruptured in the thin SiO_2_ layer [38]. Thus, when a positive voltage is applied to the top electrode, due to the migration of oxygen vacancy in the conductive filament, the filament in the SiO_2_ layer ruptures. Consequently, as shown in Figure 5c, the resistance state turns to HRS, turning the device off.

Next, the MLC characteristics of Pt/TaO_x_/SiO_2_/TaN devices are demonstrated by altering the reset voltage during the reset process. MLC characteristics are beneficial in the practical application of RRAM devices in high-density memory and neuromorphic devices [43,44,45]. It is reported that MLC behavior can be easily obtained in RRAM by controlling reset voltage and compliance current and adjusting the size of the conducting filament [46,47,48]. As shown in Figure 6a, a gradual increase in reset voltage was applied from 1.5 V to induce MLC due to the partial rupture of the conductive filament.

A gradual decrease in current level can be observed, which can be implemented as the conductive filament being partially ruptured by the increasing reset voltage and the distance between the conductive filament tips increasing, proving the existence of multi-level states in Pt/TaO_x_/SiO_2_/TaN devices. Moreover, during the reset process, when a reset voltage of 2.3 V is applied, a sudden current drop is observed. As shown in Figure 6b, after applying 2.3 V, the Pt/TaO_x_/SiO_2_/TaN device returns to its initial resistance state. We implemented a partial reset at a sweep voltage of 2 V, where its on/off ratio is about 5, and a deep reset at 2.3 V occurs where its on/off ratio is more than 10^4^. The bipolar I–V curve of the device applying deep and partial resets is shown in Figure 6c. The deep reset cell needed a higher voltage (−4 V) to change its resistance state from HRS to LRS, which can be due to the complete rupture of the conductive filament. Compared to the deep reset, the I–V of the partial reset is shown to have a more gradual change in current in the set and reset processes. A schematic illustration of the deep and partial reset processes is shown in Figure 6d. The partial reset occurs in the TaO_x_ film, leaving a strong electric field-induced localized path in the SiO_2_ thin layer. However, when a strong reset voltage (3 V) is applied, the localized filament in the SiO_2_ layer ruptures, increasing its resistance. Additional endurance and retention were measured at the read voltage of 0.3 V for both states. Both partial and deep reset states remain their resistance states for 100 cycles under DC endurance and 10^4^ s, as shown in Figure 6e,f.

The pulse trains are applied to the Pt/TaO_x_/SiO_2_/TaN device for a neuromorphic computing system. A pulse train with a set pulse (1.2 V/8 μs) as well as a reset pulse (−1.3 V/10 μs) are applied to the device to control its increase and decrease in conductance. As depicted in Figure 7a, the conductance rises by 50 consecutive set pulses and decreases by 50 consecutive reset pulses.

An abrupt rise in potentiation occurs due to the abrupt formation of conductive filaments. Also, by applying 10 consecutive identical pulse trains, its reproducibility is identified in Figure 7b. To evaluate the potentiation and depression of the device in the neuromorphic system, the Modified National Institute of Standards and Technology (MNIST) pattern recognition through a handwritten image is used. As shown in Figure 7c, the result of potentiation and depression turns into a 28 × 28 pixel handwritten image, where the clarity of the handwritten number depends on the linearity of potentiation and depression. The pattern recognition system (PRS) is composed of a deep neural network that uses handwritten image data to train and test its pattern recognition accuracy. The neural network based on Python (https://colab.google/, accessed on 15 July 2023) in Google Colab is composed of three parts: the input, hidden, and output layers. Each of the three layers is closely connected to other layers to update their parameters. Furthermore, the hidden layers are divided into three different parts, with each sub-layer consisting of 128, 64, and 32 nodes. As a result, the device has the highest accuracy of 93.56% after 10 epochs, displaying its potential to be used in synaptic applications (Figure 7d). Additionally, an ideal case potentiation and depression graph with linear and symmetric conductance changes were also inserted into the PRS system to compare it with the identical pulse application. The accuracy of the ideal case was 95.38%. Although, compared to the ideal case, some deviations in accuracy were found in the identical case, the difference between those two may be quite low. Therefore, despite its degradation, the potentiation and depression of a Pt/TaO_x_/SiO_2_/TaN device under an identical pulse application can still be used in neuromorphic computing.

Finally, spike-timing-dependent plasticity (STDP) is emulated to evaluate the capability of the device to mimic the human brain [49,50]. As shown in Figure 8a, the human synaptic neural structure can be illustrated with post- and pre-spikes, where synaptic information is delivered from pre- to post-spike.

This structure can be well mimicked in RRAM, where pre-spike and post-spike are similar to the biases on the top and bottom electrodes. Also, the process of delivering synaptic information from pre- to post-spike can be implemented through the formation of a conductive filament, allowing current flow from the top to the bottom electrode. A pulse train with a width and interval of 100 μs is applied to the top electrode (Pt) and bottom electrode (TaN) with a time interval of Δt (Δt = t_pre_ − t_post_). When the pre-spike exceeds the post-spike (Δt > 0), long-term potentiation occurs, causing device resistance to decrease. When the post-spike exceeds the pre-spike (Δt < 0), long-term depression occurs, causing device resistance to increase. Thus, by the interval of the applied pulse Δt, the conductance alters. The result of STDP is illustrated in Figure 8b. The term synaptic weight is represented as
(1)$$\Delta W=\frac{G_{f}-G_{i}}{G_{i}}\times 100\;\left(\%\right)\;$$
where *G_i_* and *G_f_* are the values of conductance in its initial state and after applying a set of pulses. The device shows a gradual change of synaptic weight (Δ*W*) and small variation over 10 cycles, proving its advantage in neuromorphic applications.

## 4. Conclusions

In this study, we demonstrate the synaptic characteristics of a Pt/TaO_x_/SiO_2_/TaN bilayer device compared to the Pt TaO_x_/TaN device. The uniformity of the resistance state is improved by inserting a 2 nm SiO_2_ layer. The conductive path can be localized between the SiO_2_ and TaO_x_ interfaces when the formation and rupture of filaments occur. The bilayer device shows more uniform resistive switching during 100 consecutive DC cycles and a longer retention time over 10^4^ s. Also, potentiation and depression are performed, and MNIST pattern recognition is evaluated based on the measurement results for the neuromorphic system. Finally, STDP is demonstrated by applying the designed pulse train to mimic the biological nerve system. Thus, by mimicking a biological synapse based on long-term memory, the TaO_x_/SiO_2_ bilayer showed favorable characteristics for future non-volatile memory applications.

## Figures and Tables

**Figure 1 materials-16-06136-f001:**
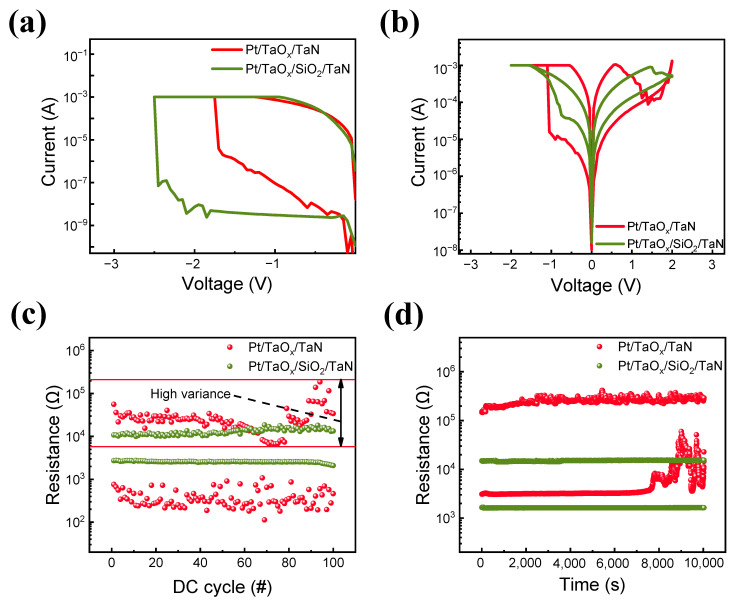
(**a**) Forming processes of Pt/TaO_x_/TaN and Pt/TaO_x_/SiO_2_/TaN devices. (**b**) I-V curves of Pt/TaO_x_/TaN and Pt/TaO_x_/SiO_2_/TaN devices for set and reset processes. (**c**) Endurance comparison over 10^2^ cycles. (**d**) Retention comparison over 10^4^ s.

**Figure 2 materials-16-06136-f002:**
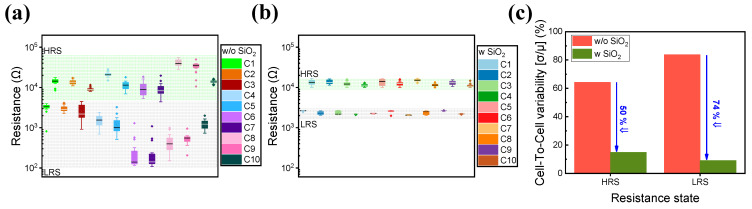
Cell-to-cell variation over 20 DC sweep cycles. (**a**) Pt/TaO_x_/TaN. (**b**) Pt/TaO_x_/SiO_2_/TaN. (**c**) Variability comparison.

**Figure 3 materials-16-06136-f003:**
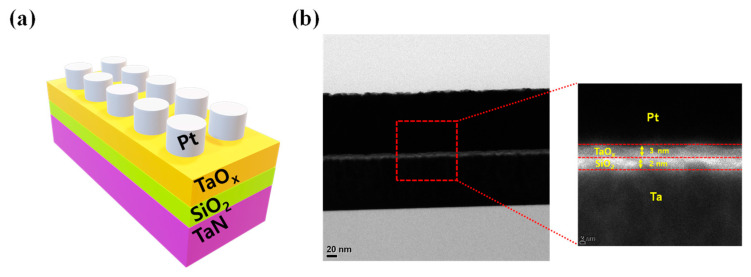
(**a**) Schematic illustration of the device structure. (**b**) TEM image of Pt/TaO_x_/SiO_2_/TaN memristor device.

**Figure 4 materials-16-06136-f004:**
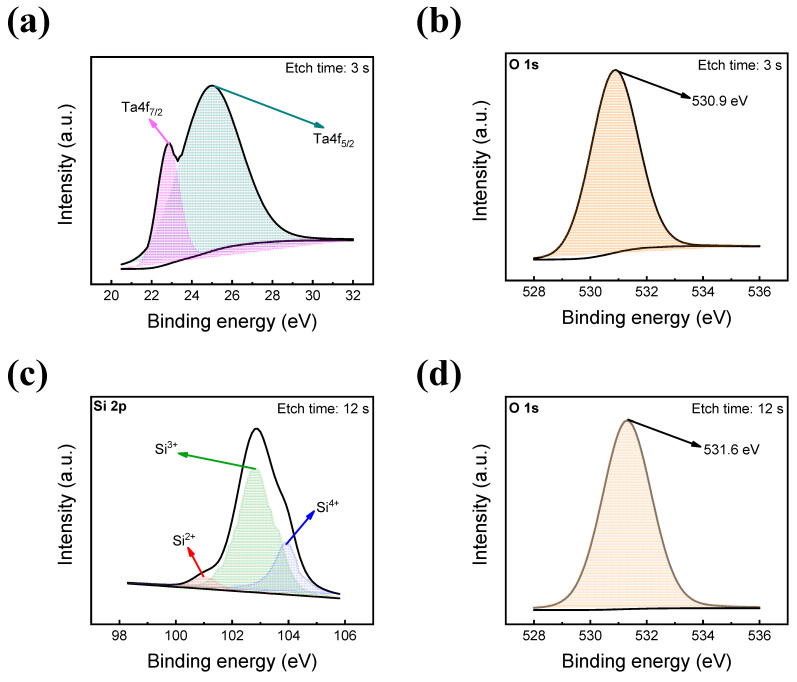
XPS characteristics of TaO_x_ and SiO_2_ thin films. (**a**) Ta 4f of TaO_x_, (**b**) O 1s of TaO_x_, (**c**) Si 2p of SiO_2_, and (**d**) O 1s of SiO_2_.

**Figure 5 materials-16-06136-f005:**
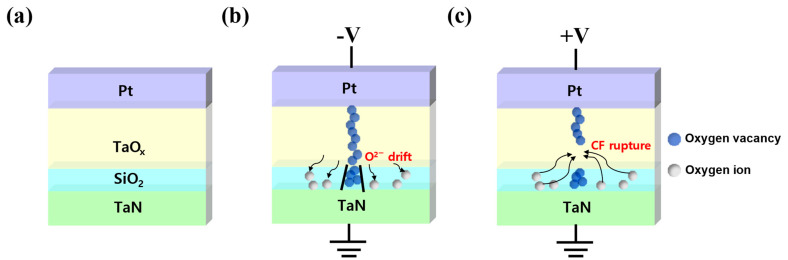
Conduction mechanism of the Pt/TaO_x_/SiO_2_/TaN device. (**a**) Initial state. (**b**) Set. (**c**) Reset.

**Figure 6 materials-16-06136-f006:**
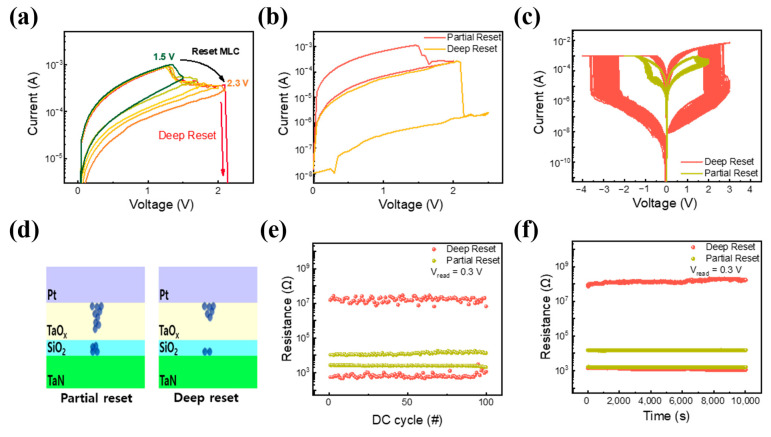
(**a**) Multi-level reset process of Pt/TaO_x_/SiO_2_/TaN. (**b**) MLC was obtained by optimizing reset voltage. (**c**) The process of inducing a partial reset to a deep reset. (**d**) Schematic illustration of deep and partial resets. (**e**) Endurance comparison of deep and partial reset over 10^2^ cycles. (**f**) Retention comparison of deep and partial resets over 10^4^ s.

**Figure 7 materials-16-06136-f007:**
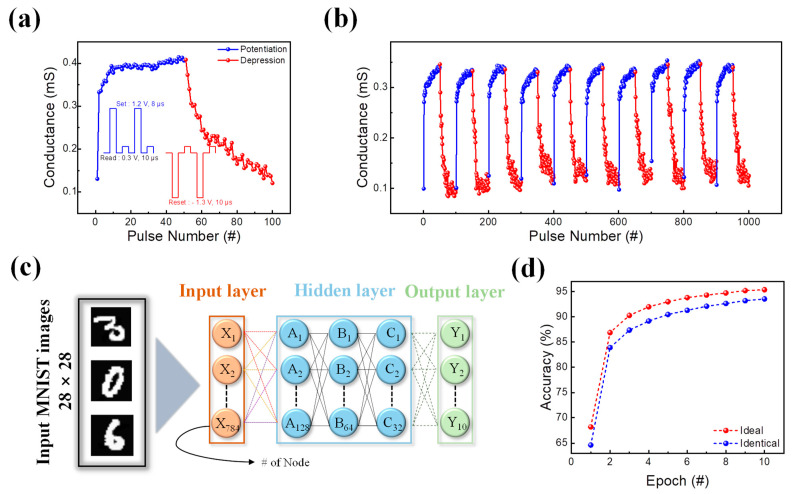
(**a**) Potentiation and depression under an identical pulse train. (**b**) 10-cycle repetition of potentiation and depression. (**c**) DNN simulation framework for MNIST pattern recognition test. (**d**) The pattern recognition accuracy of the synaptic device over 10 consecutive epochs and the accuracy comparison of ideal and identical cases.

**Figure 8 materials-16-06136-f008:**
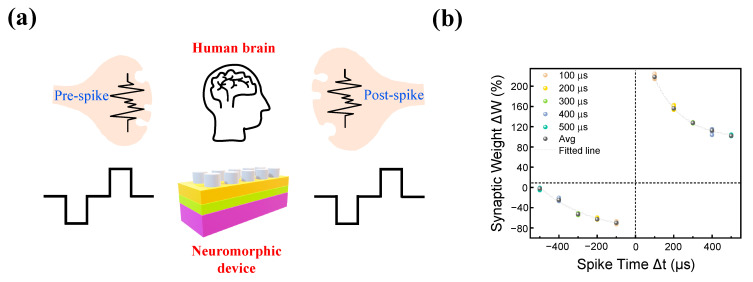
(**a**) Schematic illustration of human synaptic neural structure. (**b**) Result of STDP measurement.

## Data Availability

Not applicable.

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
