# Peer review of "Improved Uniformity of TaOx-Based Resistive Switching Memory Device by Inserting Thin SiO2 Layer for Neuromorphic System"

_materials, 2023, doi:10.3390/ma16186136_

Round 1
Reviewer 1 Report
as attached

Author Response
Dear reviewer, Thanks for your precious time and valuable comments.
Please see attached file.

Reviewer 2 Report
Please compare the accuracy and training time of your neuromorphic system on MNIST task compared to other solutions available in literature, e.g., discuss on observed results: Y. Dai, Z. Feng and Z. Wu, “A Novel Window Function Enabl es Memristor Model With High Efficiency Spiking Neural Network Applications,” IEEE Transactions on Electron Devices, vol. 69, no. 7, pp. 3667-3674, July 2022, doi: 10.1109/TED.2022.3172050. J. Li, Z. Dong, L. Luo, S. Duan, Lidan Wang, “A novel versatile window function for memristor model with application in spiking neural network,” Neurocomputing, vol. 405, pp. 239-246, September 2020, doi: 10.1016/j.neucom.2020.04.111 Please provide more details on the network you used for the MINST task, number of hidden layers, number of neurons per layer, etc. Some references are not cited within main text Please provide more details on the selection of additional layer to improve memristors response. Please discuss including appropriate experimental response whether your device belongs to volatile or non-volatile memristors. Concerning STDP, please indicate in the text which parameters of memristors is assigned to synaptic weight.I would suggest english language review
Author Response

(The authors gave the same response as above.)

Reviewer 3 Report
The authors must try to improve the content by producing consistent reasoning to support their claim.
Review comments:
1. In Figure 1 c the authors claim the resistance window reduces in case of bilayer. In Figure 6, the authors show that bilayer device also shows large resistance window and the resistance window is stable along the 100 cycles. Then Figure 1c does not make any sense. Why the authors do not consider comparing the Figure 6 (e) data with that of TaOx memory cell data? A statistical analysis between the two stacks is recommended.
2. Are these pulses (in Figure 7) optimized for obtaining LTP/LTD?
3. The sharp nonlinearity followed by a marginal modulation in the potentiation (within range of 0.3 to 0.4 mS) is challenging to use in practical architectures. Also, a linear and symmetric LTP/LTD is the best possible character. Can the authors comment on how this challenge is mitigated in the application?
4. Figure 7: On which platform the simulation was run? Are there three hidden layers? What is the activation function used for each layer? Information about the NN architecture is completely missing. What is the energy cost, if possible, to calculate?
5. The authors present a research work. But the key feature is missing. The authors mention “improvement” in the title. But the research content does not show any significant improvement. The authors should point out the what are the key features of these devices, compare the work with available literature appropriately. Similarly, a detailed analyses on “neuromorphic system”-as used in the title of the manuscript is missing.

Author Response

(The authors gave the same response as above.)

Round 2
Reviewer 1 Report
Fine with the reply. No other comments.
Author Response
Thanks
Reviewer 3 Report
In response to the Comment2, the Authors presented a response.
Figure 6 of Ref. Nanoscale Res. Lett., 2022, 17, 84 shows one type of pulse scheme and Figure 7 of the manuscript is presenting another type of pulse, if I am not wrong. The Authors may ponder on it.
2. Need to crosscheck the references. Ref. 26 is not following the format.

Author Response
Manuscript ID: materials-2557517
Title: Improved Uniformity of TaOx-based Resistive Switching Memory Device by Inserting Thin SiO2 Layer for neuromorphic system.
Authors: Dongyeol Ju, Sunghun Kim, Junwon Jang, Sungjun Kim
Reviewer #1:
Comment 1:
Figure 6 of Ref. Nanoscale Res. Lett., 2022, 17, 84 shows one type of pulse scheme and Figure 7 of the manuscript is presenting another type of pulse, if I am not wrong. The Authors may ponder on it.
Author Reply:
We are very thankful for the reviewer’s interest in our study. The potentiation and depression of the device was achieved through 50 consecutive set and reset pulse as shown in the inset of Fig. 7(a). On the other hand, for the spike-timing-dependent plasticity, a pulse scheme like Fig. 7(d) in the reference article was used to achieve timing-dependent synaptic weight change. By the change of spike-time, different resultant pulses are applied to the pre- and post-spikes, which are emulated through the top and bottom electrodes, resulting in gradual weight change.
Comment 2:
Need to crosscheck the references. Ref. 26 is not following the format.
Author Reply:
We are very thankful for the reviewer for pointing out our shortcomings and giving us a chance to improve. We fixed the reference 26 based on the format in our revised version.
Original paper:
- Yang, S.; Park, J.; Cho, Y.; Lee, Y.; Kim, S. Enhanced Resistive Switching and Synaptic Characteristics of ALD Deposited AlN-Based RRAM by Positive Soft Breakdown Process. Int. J. Mol. Sci. 2022, 23, 13249. https://doi.org/10.3390/ijms232113249
Revised paper:
- Yang, J. Park, Y. Cho, Y. Lee and S. Kim, Int. J. Mol. Sci., 2022, 23, 13249